# Touch Sensing for Robots: What is the Next?

*Abstract*—**This paper discusses the current development of research in touch sensing for robots, the major challenges, and the possible directions for future research. In the paper, I will briefly introduce the mechanism of humans' touch perception, which in years has provided lots of inspiration and guidance for roboticists to design tactile sensing frameworks. Apparently, the humans touch sensing capability is very powerful, mostly because it works as a system that integrates dynamic and static pressure sensing, kinesthetic perception, and the motion of limbs. Similarly, an intelligent touch sensing system for robots should also work as a *system*, which contains multi-modal sensory input and integration with the robot's motion system. The questions are in both *how* to build the system, and *what* is the needed system.**

## I. Introduction

Touch is an important perception modality for humans, and is the most common way we perceive the physical world when interacting with them. We touch an object or material to estimate its properties so that to better understand and evaluate it, and we also rely on touch sensing for better grasping, holding, and changing the states of physical objects. However, so far it has been challenging for robots to intelligently apply touch sensing in a similar intelligent way as humans do. The advance of touch sensing technology for robots requires a combination of good hardware design and application, good software for signal processing, and a good design of the system that combines individual sensors with then entire robot system. There are lessons we learned from the past works, but we also expect a good development touch sensing research in a systematic way for robots.

Let us take a close look at touch sense: it is the sensing about contact. In other words, it only provides local information regarding the small area at the contact surface. This makes it very different from vision, which has been widely applied in robotics, that provides global information. However, touch sensing is more active than vision, and can potentially provide more precise information, especially about the force related measurement or properties. The active nature of touch perception also means that the touch perception is not only about the reading from sensors, but also about the process of physical contact: how are those contact generated.

In this paper, I will briefly review the current development of touch sensing for robots, including the effort in sensor design, and the robotic application in manipulation and environment perception. I will then discuss the major challenges, and propose some possible future research directions for the field. I will also introduce the basic mechanism of human's touch system, as it has provided lots of inspiration and guidance for the design of touch sensing system for robots.

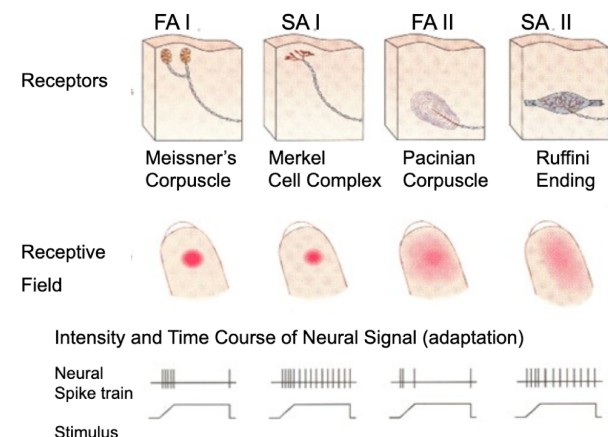

Fig. 1. An overview of skin mechanoreceptors at human fingertips[11]. There are four types of receptors buried at different depth in the skin, and they are sensitive to either dynamic or static forces.

I also would like to clarify some terminologies at the beginning of the paper. In the research fields, the most commonly seen words related to touch sensing are 'haptics' and 'tactile sensing'. If you look up the words in a dictionary, both of them mean "of or relating to the sense of touch". However, 'haptics' is mostly used for the general sensing system of touch, with most focus on kinesthetic sensing, while 'tactile' mostly refers to the force and pressure measurement at the skin. More specifically, the research field of 'haptics' grows more into the direction of studying human touch sensing and devices/systems for human-machine interface, while for robotic research people mostly use 'tactile', and sensors to measure force/pressure at the skin are mostly used in robotics for measuring touch information. I use the terminology of 'touch sensing' in this paper, because I believe for robots, the sense of touch shall be considered as a system, which includes not only the measurement of cutaneous sensing, but also other modalities such as kinesthetic sensing.

## II. Touch Sensing of Humans

For years, the study of the human touch system has inspired and guided the design of touch sensing research in robotics. The community has always wished to duplicate the strong capability of human perception and dexterity to robot systems. In this section, I will briefly introduce the touch perception system of humans, including the 'sensor hardware' part of the neurons, and the 'software' part about how human conduct touch in the behavior sense.

Humans rely on multiple types of neurons to perceive touch, and they can be divided into two groups: the neurons in the

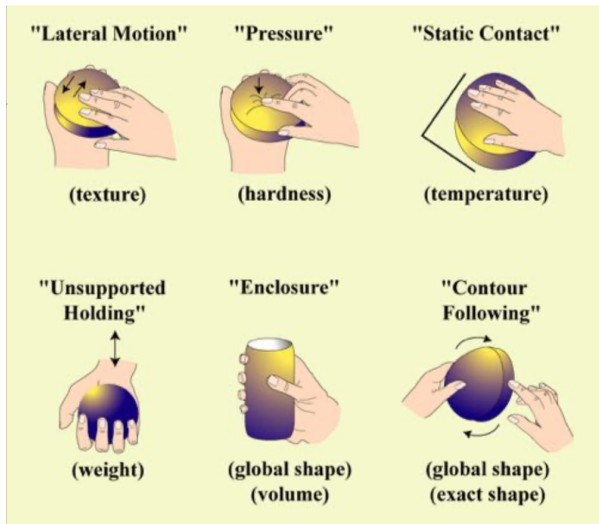

Fig. 2. Humans conduct different exploratory procedures for the aim of perceiving different properties of objects. [12]

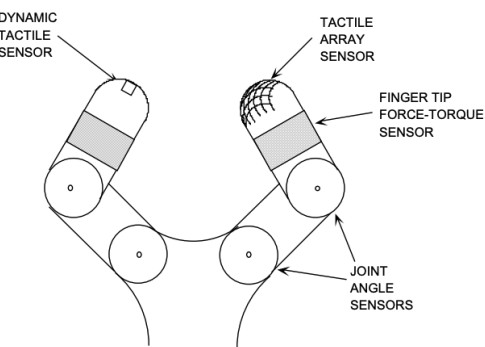

Fig. 3. The robot touch sensor system proposed by Howe [8].

skin for cutaneous sensing, and neurons in muscles, tendons, and joints for kinesthetic sensing. Within the skin, there are four kinds of mechanoreceptors to measure the force/pressure related signals, as shown in Figure 1. They differ regarding the sizes of the receptive field and the adaptation rate. They are typically referred to as FA I, FA II, SA I, SA II, where FA means 'fast adapting', SA means 'slow adapting', I refer to small receptive fields, and II refers to large receptive fields. FA receptors are sensitive to high-frequency loads in the range of 10Hz to 1000Hz, and those high-frequency signals are further magnified by the fingerprint structure on fingertips. FA receptors are also mostly used for detecting very light contact on skins and other dynamic signals such as slip, while SA receptors are mostly used for mapping pressure distribution on the skin. Those receptors are densest at the fingertips of humans, which makes the finest 2-point discrimination range of around 0.5 mm. Other than the 4 mechanoreceptors, there are other neurons within skins to measure temperature and pain. Apart from cutaneous sensing, humans also rely on kinesthetic receptors, which measure the limb positions, movement, and loads. Those receptors are embedded in muscles and joints.

Humans also apply touch sensing in an active way. As introduced by Lederman and Klatzky [12], humans have two haptic subsystems, a *sensory* subsystem and a *motor* subsystem. They work together, and the motor system serves to enhance the sensory system. A typical example is for object property perception. Lederman and Klatzky [12] summarized that humans use a set of stereotyped movement patterns that have certain characteristics while invariant for perceiving object properties, as summarized in Figure 2. Those movement patterns are named *exploratory procedures (EP)*.

As a brief summary, the touch sensing for humans works as a *system*: on the sensory part it contains multiple types of receptors that collect different types of data, and it is also combined with the motion system for active exploration

of touch. The sense we get from touch, which might be interpreted in multiple ways in our mind, like collision, light touch, slide, sticky, stiff, and etc., is a combination of signals from all types of sensory receptors and the motion subsystem. There have been lots of studies on what kind of signals is predominant in different contact scenarios, but it is certain that to achieve an intelligent and rich touch sensing capability, we humans need the integrative system.

### III. Touch sensors for robots

The sensor hardware is the base of conducting all touch-related research for robots. In this section, I will briefly review tactile sensors used on robots.

Inspired by the human's touch sensory system, [8] proposed a sensor system model for robots that works in a similar way, as shown in Figure 3. In this theory, a robot gripper uses tactile array sensor at fingertips to measure contact pressure and shapes, a dynamic tactile sensor to measure vibration, force-torque sensors near the fingertips to measure the overall force and load in the grippers, and joint angle sensors to measure the motion and configurations of the fingers. All those sensors are available, but unfortunately, it is still impossible to make a system like it is suggested. The constraints come from multiple aspects: cost, constraints in size or wiring volume, and etc.

Currently, the most commonly used touch sensors for robots are tactile sensor arrays. Those sensors measure the distribution of contact pressure at the fingertip area. Some of those sensors are commercially available, and they have a spatial resolution of 2 - 10 mm. In this case, those sensors can effectively provide information about contact force, location, and the general shape. Note that these sensors cannot measure shear forces, which is important in grasping and sliding. There have been some works on designing new structures for the sensor array so that the cell can measure both the normal force and shear force. Unfortunately, most of those designs are not used on real robots, and it's non-trivial for robotic researchers to manufacturing design.

Force-torque sensors make another category that is mature and commercially available. Existing force-torque sensors are mostly designed for robot wrists. The smallest product, the

ATI Nano-17, has a diameter of 17mm, so that it can be installed before the fingertips (an example see [15]). However, the constraint of these sensors is high cost, fragility, and large noise from both the sensor itself and the robot's motion.

One breakthrough in tactile sensor design is the emerging of the BioTac sensor [7]. The sensor aims to mimic human fingertips' multi-modal sensory capability, so that it is designed to be able to measure overall pressure, dynamic pressure, pressure distribution, and heat flow. The multi-modal measurement provides rich information about the contact, and therefore has been widely used in different robotic tasks.

In recent years, there have also been some new designs of super-human tactile sensors. For example, GelSight [23] uses optical principles to measure the geometry of the contact surface, which reaches a spatial resolution of 25 um, far beyond the resolution of human skin. Another example is the emerging of proximity sensors, which detect the existence or even the shape of the objects even before contact.

## IV. CURRENT APPLICATIONS OF TOUCH SENSING FOR ROBOTS

The application of tactile sensing in robotics can be summarized in two parts: manipulation, and exploration.

### A. Manipulation

In different manipulation tasks, the feedback from tactile sensors could help to improve the precision and robustness of manipulation. One big field of the robotic application is grasping. The tactile feedback can help robots to maintain a proper grasping force, which is neither too large to damage the objects and too small to cause grasp failure. A typical example of using tactile sensing to assist grasping is [17]. The authors mounted pressure array sensors at the fingertips of the gripper, and a force/torque sensor at the robot wrist. They manually labeled the sensor reading to the contact force and dynamic contact force, similar to the human skin's FA II and SA II. They then defined some thresholds of the contact force for maintaining a stable grasp, and use it for controlling the robot to robustly grasp and hold objects. Another trend in tactile sensing for grasping is to evaluate grasp stability using machine learning methods (example see [2]) or even re-grasp an object at better locations based on tactile readings from the current grasp (examples see [5, 3]).

In grasping research, a widely acknowledged problem is slip detection. Slip is the most common mode of grasp failure, and can hardly be detected using non-contact methods. According to high school physics, slip can be considered as the point where the shear force is larger than the product of the normal force and the friction coefficient. Therefore it is possible to detect slip by measuring the normal and shear forces respectively and compare them (example see [16]). However it is hard to apply this method broadly, because most tactile sensors cannot measure shear forces, and the measurement of exact force and friction coefficient could be challenging in real-world environments. As a lesson from humans [10], FA I is mostly used for slip detection since slipping on fingertips typically generates significant vibration. Researchers found that similarly for robots, vibration is an effective cue for indicating slip (examples see [9, 20]), and the vibration could be of physical quantities: forces, pressure, acceleration, or even acoustic input. Another way for detecting slip is to measure the micro displacement of the object surface and the sensor surface (example see [6]). But this method only works for high-resolution tactile sensors that are able to detect the micro-movement of the contact surface.

Another typical use of tactile sensing in manipulation is tactile servoing. This is by setting the manipulation goal as a certain pattern of tactile features, and train a policy for robots to reach the goal feature from the current reading. The features could be related to the geometry of contact, or the force of contact. And the policy could be classical ones like PID (example sees [13]) or deep neural networks (example sees [21]). The framework could be used in many tasks, such as contour following, surface following, and in-hand manipulation.

### B. Exploration

I use the term *exploration* to denote the application for robots to perceive objects. The perception goal could be identifying an object/material according to its shape or textures, or goes further by quantifying the properties of the objects. The object and material identification problem are typically defined as classification problems: people extract some features from the raw reading of the tactile data, and train a classifier of the tactile features. Shapes are the most used measurement for object identification. This could be about using the global shape of the objects by conducting multiple touches on the objects, and using the 3D location of the contact point to model the global shape (an example see [1]), or using the local shape from a single shot of the tactile sensor array (an example see [18]). For texture recognition, a traditional way is to use the vibration signals captured by making a robot sliding on the material (an example see [19]), but the emerging of high-resolution tactile sensors also made it possible to recognize textures directly from the spatial patterns (see [14]).

Some other works aim at perceiving more information about objects and materials, by estimating the properties of the objects. In this case, the perception of the objects is more semantically relevant, and can easily generalized to more objects. The property estimation could be about quantifying some clearly defined properties, such as hardness (example see [22]), or overall description of multiple properties of objects (example see [4]). Those works use the tactile signals a robot obtains when following one or multiple predefined exploratory procedures to obtain tactile data, and those procedures will generate different physical interactions between the robot and the target object. The result of physical interaction could be measured as forces, pressure, vibration, or deformation, and then researchers built different models to classify or regress the descriptions of the object properties from those physical measurements.

## C. Challenges

We should admit that the research of robotic touch sensing is not growing as fast as expected by many people. This is because of some major challenges in the field:

**1. Hardware.** The difficulty of accessing proper sensor hardware has been and likely will be a long-standing challenge for the field, and the challenge comes from multiple aspects. Firstly, commercialized tactile sensors are expensive, and there are not many choices. This makes the research hard to start. There have been many new sensors designed in research labs, but it is hard to replicate those sensors. Even it is possible, it takes lots of manual labor to reproduce and integrate the sensors, and the robustness of manually made sensors is mostly questionable. Secondly, most existing tactile sensors are limited in specifics, so that the signals could be either unable to provide sufficient measurement, or produce a very noisy signal. For examples, some sensors cannot provide shear force measurement, and some sensors do not have enough spatial or temporal resolution.

Lastly, the variance in sensors also slows down the communication within the community. Research on robot tactile sensing highly relies on the sensor, since all sensors have their own specifications and noise models. Therefore, the framework designed for the specific task, and the final performance, is highly customized according to the sensor, and it is hard to reproduce on new hardware platforms. This makes it difficult to evaluate the system and algorithms, as well as continue developing the method based on previous work.

**2. Disconnection between communities.** Robotic touch sensing is a highly interdisciplinary field. Designing sensors requires knowledge in electronic science, material science, microelectronics design and fabrication, optics, etc.; while using the sensors on robots requires knowledge in signal processing, robotics, and machine learning. There have been many works from the hardware communities on designing tactile sensors, but most of those sensors stopped at the lab prototype level, without the real application in robotics; and there have been many works from the robotics community or the machine learning community on using the sensors, but they are also constrained by the sensors available. Some long-standing groups that successfully worked in the field by many years, and develop the capability of both designing/making sensors and using sensors. However, I believe a healthy development of the field requires more collaboration between the two or multiple communities and taking advantage of the new technologies from different communities.

**3. System design.** As introduced in Section II, the touch sensing for humans is achieved by a system, which contains multiple kinds of sensory receptors, and the motion system. However, it is challenging to make a similar system for robots, because of the difficulty in hardware design and motion. Theoretically, it is possible to design individual sensors that measure force, pressure, vibration or load, but it is nontrivial to build a system that integrates all those sensors at a robot end-effector, and ensure they work properly. At the same time, compared to humans, current robots can hardly achieve a similar level of dexterous in motion, especially for fingers. Therefore it is harder to integrate motion with tactile sensing in different tasks.

## V. FUTURE DIRECTIONS

As discussed in Section IV-C, the development of robot touch sensing will largely rely on substantial improvement in sensor hardware development. At the same time, the research community can also benefit from the advance in algorithms and signal processing technologies. For example, in recent years, deep learning technologies, especially the ones developed for computer vision, have been applied in tactile perception and enabled complicated tactile tasks. However, in my personal view, the major research question for the next step is building a touch system for robots. We have inspiration from the design of the human haptic system, but the system of robots should be different according to the characteristics of robots. In fact, the research question is not only *how* to build the system, but also *what* the system should be like. We need more knowledge about what is needed to build the system, and we also need to compromise with what is available.

**1. Designing the robot touch system.** Similar to that of humans, the system for robot touch should contain multi-modal sensors for contact detection, and also the system of motion. The core elements for touch sensing: pressure distribution/contact geometry measurement, 3-axis force sensing, and vibration. Some other modalities could be useful too, such as proximity sensing and temperature sensing. Other than figuring out *what*, the *how* question is on how to integrate the sensors on the hardware level as well as the software level, so that to reach the full potential of the multi-modal sensory input. In addition, we should also explore how to integrate tactile reading with other parts of the robot, such as vision and audio, as well as motion planning.

An interesting thing is that although for many years, researchers in robotic tactile sensing have been trying to use human's touch system as guidance for helping robots to touch, and have been continuously discouraged by the fact that robots are not as smart as humans. However, we should note that robots run in a different way from humans, and there should be a different optimal design of the robot tactile system compared to humans. In recent years we have already witnessed some new sensor designs that exceed human sensory systems, such as a much higher spatial resolution for contact detection, and proximity sensors that are not available for humans. These indicate there could be a different design of the robot touch system, and it remains to be studied.

**2. Go dexterous.** Motion is always an important part of touch, both for tactile-based manipulation and exploration. However, for humans, those motion could be very dexterous, because of the dexterous structure of our hands. We touch objects in different ways, such as pressing, knocking, sliding, rubbing, etc. We also explore the unknown environment freely because we feel a touch from all over the fingertips, and our hand can easily apply to the expected or unexpected contact with the

environment. In comparison, most of the existing platforms for robot touch research enables only normal contact, or very limited sliding between the fingertips and an external object. Yet there is a lot to explore if we could leverage tactile sensing on multi-fingered dexterous hands, where the robot can conduct multiple contact points and generate exploratory motions in different directions.

**3. Dexterous manipulation.** I believe touch feedback will be able to play a more important role in manipulation, especially in dexterous manipulation with multi-fingered hands, which has been a challenge in manipulation. By embedding proper touch sensors, a robot should be able to perceive complicated contact situations in manipulation cases, and therefore enables dexterous manipulation in real-world scenarios.

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
