# OpenReview forum: "Touch Sensing for Robots: What is the Next?"
_roboticsfoundation.org/RSS/2020/Workshop/RobRetro — RobRetro 2020_

### Official Review · AnonReviewer1 · 2020-06-24
**Overview of Touch Sensing Technologies and Challenges**

**Confidence:** 4
**Rating:** 8

**Review:**

This work provides an overview on touch sensing from multiple angles: Human touch, touch sensors for robotics, how to use touch for manipulation, current challenges and the authors opinions on future directions. While this paper reads a bit more like a survey, it does make a good retrospective on touch sensing. Especially the section on Challenges the authors discusses their opinions on why touch sensing is not developing as fast as some people seem to expect.

Some considerations/comments for the final version:
A paper on a novel sensor you might want to add to your overview: https://ieeexplore.ieee.org/abstract/document/9018215

---

### Decision · Program_Chairs · 2020-06-25

Accept